# Therapeutic Manipulation of Macrophages Using Nanotechnological Approaches for the Treatment of Osteoarthritis

**DOI:** 10.3390/nano10081562

**Published:** 2020-08-09

**Authors:** Aldo Ummarino, Francesco Manlio Gambaro, Elizaveta Kon, Fernando Torres Andón

**Affiliations:** 1Department of Biomedical Sciences, Humanitas University, Via Rita Levi Montalcini 4, 20090 Pieve Emanuele, Milan, Italy; aldo.ummarino@hunimed.eu (A.U.); francescomanlio.gambaro@st.hunimed.eu (F.M.G.); 2IRCCS Istituto Clinico Humanitas, Via A. Manzoni 56, 20089 Rozzano, Milan, Italy; 3Center for Research in Molecular Medicine & Chronic Diseases (CIMUS), Universidade de Santiago de Compostela, Campus Vida, 15706 Santiago de Compostela, Spain

**Keywords:** nanomaterial, macrophage, nanoparticle, drug delivery, immune system, anti-inflammatory, innate immunity, osteoarthritis

## Abstract

Osteoarthritis (OA) is the most common joint pathology causing severe pain and disability. Macrophages play a central role in the pathogenesis of OA. In the joint microenvironment, macrophages with an M1-like pro-inflammatory phenotype induce chronic inflammation and joint destruction, and they have been correlated with the development and progression of the disease, while the M2-like anti-inflammatory macrophages support the recovery of the disease, promoting tissue repair and the resolution of inflammation. Nowadays, the treatment of OA in the clinic relies on systemic and/or intra-articular administration of anti-inflammatory and pain relief drugs, as well as surgical interventions for the severe cases (i.e., meniscectomy). The disadvantages of the pharmacological therapy are related to the chronic nature of the disease, requiring prolonged treatments, and to the particular location of the pathology in joint tissues, which are separated anatomical compartments with difficult access for the drugs. To overcome these challenges, nanotechnological approaches have been investigated to improve the delivery of drugs toward macrophages into the diseased joint. This strategy may offer advantages by reducing off-target toxicities and improving long-term therapeutic efficacy. In this review, we describe the nanomaterial-based approaches designed so far to directly or indirectly manipulate macrophages for the treatment of osteoarthritis.

## 1. Introduction

Osteoarthritis (OA) is the most common joint pathology, affecting approximately 33% of the population above 65 years of age with a predilection for the female gender [1]. This disease is able to potentially affect every joint of the human body, but most commonly affecting the knee (85% of the worldwide burden of OA) [2], followed by the hip and the hand joints [3]. The progression of OA pathology results in pain and functional disability, which impact severely on patient’s quality of life, making this condition expected to be the 4th leading cause of chronic disability worldwide by the year 2020 [4].

Macrophages are key regulators of OA physiopathology. These innate immune cells trigger the inflammatory response in the joint microenvironment through the secretion of cytokines and other molecular mediators, control the activity of the adaptive immune system, and also influence other cells such as chondrocytes [5]. Macrophages are characterized by a high plasticity dictated by their continuous adaptation and response to specific local stimuli [6]. At the edges of this continuum polarization status of macrophages, two extreme phenotypes can be defined, which have been characterized in terms of gene expression, the pattern of surface molecules, and the production of biological mediators and metabolites [7]. The presence of lipopolysaccharide (LPS), interferon gamma (IFN-γ), and tumor necrosis factor alpha (TNF-α) induces the polarization of macrophages toward an M1-like phenotype, presenting pro-inflammatory functions with the release of interleukin (IL)-1β, IL-6, IL-12, IL-23, and TNF-α [8]. Conversely, their exposure to IL-4 and IL-13 polarizes macrophages toward an M2-like phenotype, which confers to these cells an anti-inflammatory behavior, secreting IL-10 and TGF-β, among others [9]. In OA joints, an imbalance between M1 and M2 macrophages has been observed, with an increase of the first subset that accelerates the onset of the pathology and the related symptoms [10]. Taking into account this knowledge, the reprogramming of M1-like macrophages toward M2-like anti-inflammatory effectors appears to be a reasonable strategy for the treatment of OA-related inflammation.

Most frequently, drugs, such as non-steroidal anti-inflammatory drugs (NSAIDs) or corticosteroids, but also other molecular approaches, for example cytokine-blocking antibodies, anti-inflammatory peptides, or even hyaluronic acid, have been given systemically or locally injected in the joints to ameliorate inflammation [11,12]. Several investigations have studied the pros and cons of different treatments and routes of administration. Systemic (oral) administration is typically used in the clinic, although it is discouraged due to the little amount of drug administered that effectively reaches its target diseased tissue, as well as adverse effects of anti-inflammatory compounds in off-target organs [13]. The intra-articular administration of drugs locally, mainly corticosteroids [14], has been applied in the clinic and also explored in basic research with different methodological approaches [15]. However, this strategy comes also with several obstacles, such as short-term efficacy, in part due to lymphatic and blood drainage of the drug in less than 2 h from the joint [16], and uncomfortable application, typically requiring multiple injections. Specific macrophage targeting has been barely explored, although this strategy may have the advantage to reduce the amount of drug reaching other cells into the joint, such as chondrocytes or fibroblasts.

Nanotechnological approaches offer a wide range of possibilities to overcome these issues, as different nanomaterials can be now engineered to load and deliver therapeutic molecules in a controlled release manner and even help the drug reach specific cell populations, such as macrophages. Indeed, the phagocytic nature of macrophages has been demonstrated to significantly favor the uptake and accumulation of nanostructures, at inflamed sites, inside these immune cells versus other cellular or extracellular compartments [17,18,19]. In the last decade, numerous types of nanocarriers have been engineered, such as nanoparticles, liposomes, or hybrid nanosystems to improve the delivery of drugs toward macrophages in the diseased joints (Figure 1). Interestingly, also some nanostructures have been designed to prevent macrophage uptake to slow down the drug consumption and prolong the therapeutic effect [20,21]. In this manuscript, we review the nano-based drug delivery strategies aimed to manipulate the immune system in the context of OA, with a particular focus on those designed to specifically target and reprogram macrophages.

## 2. Pathophysiology of Osteoarthritis and Role of Macrophages

Osteoarthritis can be classified, according to its leading causes, into primary or secondary based on the underlying etiology. The primary, also known as idiopathic, form of OA presents a complex multifactorial pathophysiology, which can be partially explained by the combination of the following factors: obesity, aging, limbs malalignment, excessive joint loading, and/or genetics. On the other hand, secondary OA is defined by the occurrence of a recognizable causative agent, such as a trauma, joint malformation at birth, rheumatoid arthritis, or surgery (i.e., post-meniscectomy [22]).

In addition to these local triggers of OA, systemic inflammation (as in the case of metabolic syndrome) may play a role in the development of the disease through the establishment of a chronic, systemic low-grade inflammation and the release of circulating adipokines, such as leptin and adiponectin [23]. In particular, leptin in the joints has been shown to increase the expression of inflammatory mediators, such as inducible Nitric oxide synthase (iNOS), cyclooxygenase 2 (COX2), Prostaglandin E2 (PGE2), IL-6, and IL-8 in the cartilage [24]. Meanwhile, adiponectin, presenting lower levels in obesity [25], has a protective role in the articulations by upregulating in chondrocytes the expression of tissue inhibitor metalloproteinase 2 (TIMP2) and IL-10, and preventing cartilage deterioration by the downregulation of matrix metalloproteinase 3 (MMP-3) [26]. In addition, the diagnosis of diabetes mellitus type 2 has emerged as an independent risk predictor of OA severity and arthroplasty requirement [27], possibly owing to a direct detrimental effect of hyperglycemia and chronic low-grade inflammation on cartilage metabolism [28]. Despite the mechanism responsible for the disease progression, OA commonly arises showing similar pathological alterations, as follows: the first joint compartment to be affected is the articular cartilage and then progressively, the disease involves every component of the joint, as the subchondral bone and the synovium, making OA a “whole joint disease” [29].

Increasing basic and clinical experimentations has demonstrated a key role for macrophages in the development and progression of OA [30,31]. The principal involvement of this innate immune cell population has been observed in patients by single-photon emission computed tomography together with conventional tomography (SPECT-CT) in vivo, revealing that 76% of joints affected by OA present activated macrophages and that the number of these cells correlates with radiographic joint disease severity and symptoms [32]. Interestingly, an increased number of activated macrophages is commonly present already at the early stages of OA, suggesting their primary role in the disease development, since the initial phase [33]. The fact that higher macrophage infiltration is observed in early OA, rather than in late OA, further strengthens the concept that these cells are involved from the initial phases of the disease [34]. In addition to their number, the functional polarization of macrophages, i.e., toward an M1-like pro-inflammatory phenotype, plays a fundamental role in the initiation, progression, and resolution of osteoarthritis, also in the knee, as described below.

Physiologically, in the context of the knee, the resident macrophages are present inside the synovial membrane, maintaining their numbers thanks to a pool of locally proliferating mononuclear cells, which are embedded into the synovial tissue and exert a homeostatic function [35]. Whenever these resident monocytic cells detect the presence of exogenous or endogenous pro-inflammatory stimuli, a chain of events known as inflammatory cascade ensues. In the case of OA, acute trauma or chronic overuse of the joint results in tissue injury that is associated with the production of damage-associated molecular patterns (DAMPs), such as fibronectin [36] and oligosaccharides from degraded hyaluronan [37]. These molecules are able to activate resident synovial macrophages toward an M1-like phenotype, by binding to pattern recognition receptors (PRRs) and influencing downstream transcription factors such as the nuclear factor kappa-light-chain-enhancer of activated B cells (NF-κB), which support and boost the pro-inflammatory cascade. Once activated, synovial macrophages exert, directly or indirectly (i.e., stimulating other cells), several actions in the joint microenvironment. Macrophages are responsible for amplification of the inflammatory response, direct tissue injury, and orchestration of other synovial cells. The first action is due to the secretion of signaling molecules, such as pro-inflammatory cytokines (IL-1β, TNF-α, IL-6, IL-12, and IL-23) and chemokines (CCL-2 and CXCL-8) that lead to further leukocyte recruitment, including monocytes. In this way, newly recruited monocytes are differentiated and polarized toward pro-inflammatory macrophages, which join the activated resident macrophages [35]. This accumulation of leukocytes in the OA synovia eventually results in its thickening (a key morphological feature of synovitis) and the perpetration of synovial inflammation. In this context, macrophages can also exert the direct injury of the tissue through the secretion of matrix metalloproteinases (MMP1, MMP3, MMP13), nitric oxide (NO), and reactive oxygen species (ROS) [38,39], leading to the release of more DAMPs and the initiation of a positive feedback loop. Furthermore, the ability of macrophages to orchestrate surrounding cell populations has also been demonstrated. Macrophages can directly inhibit chondrogenesis in mesenchymal stem cells (MSCs), as indicated by decreased expression levels of genes involved in collagen type 2 (COL2) and aggrecan (ACAN) synthesis, which are two fundamental constituents of human cartilage responsible for the resilience of this tissue [40]. The aforementioned actions are exploited mainly by M1-like pro-inflammatory macrophages [41], and their pathological role was confirmed by the selective removal of macrophage-like synovial cells that resulted in the downregulation of pro-inflammatory cytokines and MMP produced by fibroblasts [42].

On the contrary, also in the context of knee OA, it was observed that macrophage polarization toward an M2-like anti-inflammatory phenotype promotes cartilage repair by expressing pro-chondrogenic genes, such as TGF-β and insulin-like growth factor (IGF), resulting in an increased synthesis of collagen type II and glycosaminoglycan, which is accompanied by the inhibition of chondrocytes’ apoptosis [43]. These observations are in line with the theoretical classification of M1-classically activated macrophages with a pro-inflammatory behavior versus the M2-alternatively activated cells with anti-inflammatory properties [44] (Figure 2). Furthermore, it was reported that the imbalance between M1 and M2 macrophages in favor of the former is associated with the severity level of osteoarthritis [10].

Overall, given the key role played by macrophages in the pathogenesis of OA, it becomes evident that their therapeutic manipulation represents an important target for the treatment of the disease. Some drugs with ability to decrease the M1/M2 macrophage ratio, as desired, for the resolution of knee OA already exist [46,47,48]. However, the major challenge, still not achieved, consists in the appropriate spatio-temporal manipulation of the immune response in the joint, which may be achieved by a focused therapeutic intervention of macrophages. For this, we believe that the development of new drug delivery approaches is needed, such as administration tools guaranteeing a drug concentration inside the joint at the right dose and for a period of time sufficient to exert a satisfactory long-lasting therapeutic effect. In the next section, we describe recent investigations in the field of nanotechnology aiming to control the delivery of anti-inflammatory drugs toward macrophages in the context of OA.

## 3. Pharmacological Manipulation of Macrophages for the Actual Treatment of OA

The pharmacological treatment of OA, according to the latest Osteoarthritis Research Society International (OARSI) guidelines [49], is commonly addressed with one or two drugs that are administered either locally (topical or intra-articular) or systemically. To assess the efficacy of the pharmacological treatments, clinicians usually refer to a score established by the OARSI that measures the joint damage in a histopathological manner. This score relies on two parameters: the grade and the stage. The grade is a measure of the vertical extent of the damage of the joint, as this is a reliable indicator of the severity of the disease. On the other hand, the stage is a horizontal measure of the surface/area/volume that is affected by OA. The final OARSI score is obtained by multiplying the grade for the stage, and it allows following the therapeutic activity in a quantitative manner [50].

The drugs currently approved for OA that perform their action at least in part by direct or indirect manipulation of macrophages can be classified into the following main categories: non-selective non-steroidal anti-inflammatory drugs (NSAIDs), NSAIDs inhibiting selectively cyclooxygenase (COX)-2 (coxibs), corticosteroids, and/or hyaluronic acid. Other pharmacological strategies, such as the use of mAbs or nucleic acids are nowadays the object of intensive investigations, but they have not yet entered the clinic [48,51,52].

Non-steroidal anti-inflammatory drugs (NSAIDs), such as aspirin, ibuprofen, diclofenac, or naproxen, exert their anti-inflammatory effect by inhibiting cyclooxygenase (COX) enzymes (both the isoform COX-1 and COX-2) expressed by macrophages [53,54] that metabolize arachidonic acid into thromboxanes and prostaglandins involved in inflammation. Due to their broad effect, non-selective NSAIDs have shown a series of harmful effects on the gastrointestinal tract [55] and kidneys’ functions [56]. To overcome these issues, semi-selective NSAIDs (i.e., indomethacin, diclofenac, meloxicam) with a higher affinity for COX-2 versus COX-1, and coxibs (i.e., celecoxib, etoricoxib, parecoxib) specifically targeting COX-2 enzymes have been introduced in the clinical practice. Unfortunately, the systemic administration of COX-2 inhibitors showed also harmful side effects in patients with a known cardiovascular risk [57]. NSAIDs are commonly used, either locally or systemically, for OA. Although the oral administration is always advantageous for the possibility of self-administration, ease of ingestion, and pain avoidance, the local approach is preferred to achieve an intra-articular efficacy and to minimize the adverse events described above.

Corticosteroids are a class of steroid hormones with a powerful anti-inflammatory effect that is mediated by their binding to the glucocorticoid receptor, which is activated and translocated to the nucleus of immune cells, activating the synthesis of lipocortin-1 and gluconeogenesis, among others [58]. Through this mechanism, they modulate a wide range of physiological processes, including stress and immune response. In the clinic, prednisolone, dexamethasone, and others represent well-established treatment options for several rheumatological and orthopaedical pathological conditions, including knee osteoarthritis [59,60,61]. Their use in OA is usually restricted to intra-articular injections for short periods, as prolonged treatment with corticosteroids can induce extreme immune suppression, which avoids the resolution of the inflammation [62] and/or bone resorption in the joint, by impairment of osteoblasts and osteoclasts recruitment [63].

Although not classified as a drug, the intra-articular administration of hyaluronic acid (HA), a glycosaminoglycan widely distributed throughout the body in several forms related to different molecular weights, has shown relative efficacy in some cases of OA. While low molecular weight HA presents pro-inflammatory effects mainly acting on toll-like receptor-2 (TLR-2) and toll-like receptor-4 (TLR-4) receptors on macrophages [64]; on the contrary, physiologically produced or therapeutically injected high molecular weight hyaluronic acid (HMW-HA) has demonstrated anti-inflammatory effects on macrophages [65] in different settings [66,67,68]. The local injection of HMW-HA in the knee of patients has demonstrated a significant improvement in OA symptoms [66]. Furthermore, HA is widely applied in dentistry [69], regenerative medicine [70,71], and ophthalmology [72]; and it is gradually becoming a useful therapeutic tool for the intervention of other pathological conditions [73].

Finally, monoclonal antibodies-based therapies are increasingly being used in randomized clinical trials, presenting mixed results in some cases. For example, the weekly subcutaneous injection of adalimumab, an antibody that interferes with TNF-α signaling, which is one of the main pro-inflammatory pathways in macrophages [74], did not result in a positive effect in OA management versus placebo in patients with erosive hand osteoarthritis (HUMOR trial [75]). On the contrary, the injection of 10 mg of the same antibody in the knee joint of patients with moderate/severe knee OA was effective and well tolerated [76]. This finding was also reported in a separate clinical trial for the intra-articular injection of etanercept, which is a fusion protein produced by recombinant DNA used to inhibit the TNF-α receptor [77]. Promising results had been observed by nerve growth factor (NGF) inhibition, which is produced also by macrophages [78], using fasinumab antibodies to prevent OA pain [79]. Subcutaneous injection of fasinumab improved knee/hip OA pain and function [80]. Several ongoing clinical trials are evaluating monoclonal antibodies to inhibit other macrophagic pro-inflammatory pathways such as anti-IL-1β or anti-IL-6 signaling. Lutikizumab, anti-IL-1α/β, resulted in a limited improvement in patients with knee OA [81], while no clear evidence is available for IL-6 blockade, despite its common application in rheumatoid arthritis [82].

Among the pharmacological approaches presented, the systemic or local administration of NSAIDs is given to most of the patients with OA, whilst intra-articular corticosteroids and hyaluronan are conditionally recommended only for the management of particular cases of knee OA. However, the local application of these drugs is able to reduce the inflammation only upon administration, and they are rapidly cleared from the joint by blood and/or lymphatic drainage [83], thus restricting the long-term efficacy of the therapy at the desired target location. For this reason, the oral administration of non-selective NSAIDs or Coxibs, despite their side effects [55,57], is usually prescribed in high doses for extended periods of time to almost all the patients with knee OA. Interestingly, it has been observed that a dose of NSAIDs 100–1000-fold higher than the one used to reduce inflammation could be also beneficial to inhibit the apoptosis of chondrocytes and prevent their dedifferentiation [84]. As it would be dangerous to administer such a dose orally, further efforts to optimize intra-articular delivery are highly desirable. For this, new drug delivery systems are being explored, aiming to implement the local treatment with high and sustained therapeutic efficacy on macrophages in the joint, by controlling the dose and time of release of specific drugs at the right location, while avoiding their systemic biodistribution and consequent adverse effects. The optimal drug delivery approach into the joint must allow a unique implantation, avoiding also adverse effects connected with the practice of multiple injections, such as mild swelling [85] or infection [86] of the joint. In the next section, we present the most relevant nanomaterial-based approaches, which have been investigated so far for this purpose, manipulating macrophage behavior in osteoarthritis.

## 4. Nanomaterial-Based Approaches for Macrophage Manipulation in OA

The therapeutic manipulation of macrophages using nanotechnology and other drug delivery systems has been usually approached taking into consideration the high ability of these cells to recognize and phagocyte “non-self” or “potentially” dangerous pathogens. The physicochemical properties of nanoparticles (NPs), such as size or surface charge, or even the surface functionalization of nanomaterials can be designed to favor their non-specific [87,88,89] or specific, receptor-mediated uptake by macrophages [90,91]. The decoration of NPs with specific ligands, such as sugars, lipids, peptides, or even antibodies, has showed improved macrophage targeting [92]. Some of these ligands have been implemented to target macrophages in other pathologies, i.e., cancer, atherosclerosis, or infectious diseases, and they have been reviewed elsewhere [19,90,91,93]. Although out of the scope of this review, these investigations could provide important information for NP–macrophage targeting in OA.

Upon recognition and internalization by macrophages, these NPs are commonly disassembled inside the lysosomes or cytosol of the cells, allowing drug release. As a contrary approach, some biomaterial-based approaches have been designed with non-spherical sizes or enveloped into protective coatings (i.e., PEG or erythrocyte membranes), to avoid or delay macrophage recognition [94,95]. This strategy has been used in the context of intra-articular delivery to achieve a long-lasting release of the drug from the biomaterial, which is not rapidly internalized by macrophages [20,21]. Finally, some nanotechnology-based approaches have been designed to release the drug extracellularly, not inside macrophages, or even using selected ligands to target other cellular components of the joint (i.e., chondrocytes, synoviocytes, fibroblasts), but always aiming to achieve an anti-inflammatory effect.

The majority of the nanomaterial-based approaches designed to inhibit macrophage-induced inflammation for the treatment of OA have used natural or synthetic polymers with a good regulatory profile [96]. Liposomal formulations, made up of a phospholipidic bilayers resembling cellular membranes, are the second most common group of nanosystems applied to improve the delivery of hydrophilic (loaded in the inner core) or lipophilic (trapped in the lipidic bilayer) anti-inflammatory drugs [97]. Furthermore, here we provide some examples of metallic- or carbon-based NPs and nanomaterial-based scaffolds or gels that have been engineered to control the intra-articular release of drugs in the context of OA. In all cases, for the selection of the nanomaterials, their biocompatibility, biodegradability, and safety are major premises that must be tested, in addition to the capability of the nanotechnological approach to improve the therapeutic efficacy of the free pharmacological molecule. A schematic presentation of all the nanomaterial-based approaches presented in this review manuscript is provided in Table 1.

### 4.1. NSAIDs-Loaded Polymeric Nanoparticles

NSAIDs, as the most used anti-inflammatory and pain relief drugs used worldwide, are commonly given by oral or local administration to patients with knee OA. Several polymeric-NSAID-based approaches have been investigated to improve their local efficacy and reduce their side effects. As an example, Kang et al. developed thermoresponsive polymeric nanospheres containing diclofenac (DCF) and kartogenin (KGN), a pro-chondrogenic compound. They used F127 pluronics and triblock copolymers linked to obtain an amphiphilic final structure and then emulsified with KGN molecules conjugated to the natural polymer chitosan (CS), allowing the binding of KGN/CS molecules on the hydrophilic part of the F127. Subsequently, diclofenac was loaded into the inner hydrophobic core of the NP. These nanospheres showed, in vitro, the individual and sustained drug release of the two drugs, KGN and DCF, which can be accelerated when exposed to cold temperature (4 °C). This spontaneous drug release, subsequent to the cold temperature, is achieved by the enlargement of the F127 segments, generating a loose structure that is more prone to undergo hydrolysis with the consequent release of KGN. In parallel, the increased water permeability occurring at low temperature allows a rapid burst release (12 h) of DCF from the lipidic core.

These nanospheres prevented the secretion of interleukin-6 (IL-6) from LPS-treated U937 cells. Despite these positive results using U937 as an in vitro model of human monocytes from tumoral origin, similar experiments with primary macrophages would be preferred for better mimicking the clinical situation. In rat models of OA induced by anterior cruciate ligament transection (ACLT) and destabilization of the medial meniscus (DMM), these nanospheres showed a satisfactory reduction of the OARSI score [98]. In another study, self-assembling nanosystems made up of Poloxamer 407 (highly hydrophilic) and Tetronic 90R4 (intermediate hydrophilic) were loaded with indomethacin. In addition to these amphiphilic polymers with surfactant properties, also poly (lactic-co-glycolic acid) (PLGA) and articular proteoglycans (collagen, gelatin, and glucosamine HCl) were added in different ratios to achieve a controlled and sustained drug release. The best prototypes, based on their drug-retaining capacity, were investigated in arthritis models consisting of male albino rats injected in their knees with the antigens ovalbumin and complete Freund’s adjuvant. After 2 intra-articular injections, the indomethacin nanosystems showed significant therapeutic efficacy in terms of knees’ histopathological features and TNF-α concentration in serum. The same dose of the free drug was not effective, which is probably due to its poor solubility in the synovial fluid and is not a problem in the case of the nanosystems [84]. Piroxicam is another NSAID used for the long-term therapy of joints’ inflammatory diseases, which has been encapsulated for the intra-articular delivery using NPs with different surface charges [99]. In this case, the NPs composed of PLGA and Eudragit RL, a positive charged polymer intended to interact with negative charged HA, were investigated to reduce the efflux of NPs from the joint. As expected, upon intra-articular injection in healthy rats, the cationic NPs showed a higher retention of piroxicam in the synovial fluid versus the neutral NPs or the free drug, as evaluated by LC-MS/MS. Accordingly, the kinetics of piroxicam in the blood showed a delayed and minor peak for the cationic NPs group in contrast with the rapid increase observed for the free drug, thus reducing the risk of systemic adverse events [99].

Among coxibs, with the ability to inhibit specifically COX2 enzyme, celecoxib (CXB) has shown the safest profile related to the induction of cardiovascular side effects [100]. For OA, Villamagna and colleagues designed CXB-loaded NPs, based on degradable poly(ester amide)s (PEAs), with different monomers intended to modulate their thermal and mechanical behavior, and consequently the drug release. By dialysis membrane, in vitro, a slow release of only 25% of CXB at 40 days, with the drug still present in NPs at day 60, was observed. Using sheep in vivo models of OA, lameness, joint effusion, periarticular swelling, fever, heart rate, synovial fluid, and plasma were evaluated, showing satisfactory anti-inflammatory activity for the CXB NPs, with only a minor increase in white blood cells in synovial fluid and mild synovial intimal hyperplasia [101]. The sheep or equine model represents important added value, in terms of the anatomical and histological similarity of the knee to humans, but also immunological response versus murine models. The evaluation of macroscopic lesions and degrees of damage in mice is more challenging due to their small size [102]. NPs with PLGA and PEG have been also prepared to load etoricoxib, which is a drug with higher therapeutic activity at lower doses but also higher adverse effects than CXB. Sustained drug release was observed for these NPs up to 1 month, by the dialysis membrane, although showing an initial burst of 25% release in the first five days. In rats with OA, induced by the anterior cruciate ligament transection technique, the injection of etoricoxib NPs showed a lower OARSI score, no signs of inflammation, higher expression of type II collagen and aggrecan (mandatories for cartilage integrity), and lower levels of MMP-13 and A disintegrin and metalloproteinase with thrombospondin motifs 5 (ADAMTS-5), which are two proteases produced by macrophages that are responsible for joint destruction [96]. Despite the promising anti-inflammatory activity for these NSAIDs-loaded NPs, their translation to the clinic is delayed by still ongoing pre-clinical experiments and toxicity/safety studies, which must be ideally performed in different animal species i.e., rodents, sheep, and/or equine models.

### 4.2. Corticosteroid-Loaded Polymeric Nanoparticles

With many corticosteroids currently applied in the clinical practice, dexamethasone (DX) is one of the most popular for encapsulation into NPs, thanks to its powerful anti-inflammatory activity, even at low dose [103]. The first attempts to encapsulate DX in PLGA polymeric NPs suffered from a low drug loading and a fast crystallization [104]. To overcome this impasse, dexamethasone palmitate (DXP), a prodrug of DX characterized by a high hydrophobicity, was encapsulated in poly(ethylene glycol) (PEG)ylated NPs with high efficiency (98% *w*/*w*) and sustained release in vitro up to 25 h, showing a peak of 60% of DXP release at 5 h. Using LPS-stimulated RAW 264.7 murine macrophages, these DXP NPs versus the free drug showed better inhibition of TNF-α secretion. In this case, primary macrophages may also give a more sensitive indication of the anti-inflammatory efficacy of DXP NPs than the RAW 264.7 cell line. Upon intravenous injection in a murine model of rheumatoid arthritis obtained through the intradermal injection of an emulsion of complete Freund’s adjuvant and type II collagen, DXP NPs demonstrated a higher accumulation into the inflamed joints, with lower systemic biodistribution and mild to little adverse reactions. The testing of different doses revealed the necessity of 1 mg/kg of DXP NPs to achieve a significant reduction in inflammatory histological signs [105]. In another report, betamethasone sodium phosphate (BSP) was incorporated into PLGA nanospheres to reduce the inflammation in rabbit models of ovalbumin-induced chronic synovitis. The intra-articular injection of BSP nanospheres, together with the antigen ovalbumin, in one of the two joints of the rabbits was able to reduce joint swelling up to 21 days. This prolonged therapeutic effect, not observed for the free steroid, demonstrated the comparative advantage of the nanosystem [106]. These data demonstrate the feasibility of using nanotechnological approaches for the long-term intra-articular delivery of steroids. However, due to the local harmful effects of these drugs, which are described in Section 3, we would prioritize the development of steroid-based drug delivery systems to facilitate their administration and short-/medium-term release (less than 1 month).

### 4.3. Polymeric Nanoparticles Loaded with Other Anti-Inflammatory Molecules

As an alternative to conventional anti-inflammatory drugs, another approach has considered the use of anti-inflammatory peptides, with the ability to inhibit inflammatory pathways, and their loading into NPs was explored to improve their efficacy in vivo. The cell-penetrating anti-inflammatory peptide KAFAK was loaded into hollow thermosensitive poly(N-isopropylacrylamide) (pNIPAM) with ability to reduce IL-6 secretion. These NPs present the advantage of undergoing hydrophobic collapse at physiological temperature; thus, the peptide is trapped in the core of the NP upon injection, and its release is prolonged over time [107,108]. The in vitro testing of pNIPAM-NPs showed high uptake by RAW 264.7 macrophages, and a stronger inhibition of IL-6 production than the free anti-inflammatory peptide KAFAK was observed using ex vivo models of OA, which were obtained from bovine cartilage plugs treated in vitro with trypsin to remove aggrecan and induce inflammation [109]. In another report, KAFAK-loaded pNIPAM-NPs were co-polymerized with 2-acrylamido-2-methyl-1-propanesulfonic acid (AMPS), which enhances the interaction between the NPs and the KAFAK peptide, resulting in increased drug loading. These NPs were tested in vitro using LPS pre-stimulated-THP-1 macrophages, and ex vivo using cartilage plugs from bovine knee joints pre-treated with interleukin-1β (IL-1β) to initiate inflammation. In both cases, the KAFAK NPs were able to reduce the inflammation, as determined by TNF-α production by the THP-1 cells and IL-6 secretion in the ex vivo models [110].

The strong pro-inflammatory effect of IL-1β can also be blocked by the IL-1 receptor antagonist (IL-1ra), which is a receptor naturally secreted by polymorphonuclear cells after their stimulation that has been effectively used in the clinic as a recombinant protein (Anakinra^®^) [111,112]. In the context of OA, IL-1Ra-poly(2-hydroxyethyl methacrylate)-pyridine NPs have been developed and tested in vitro on RAW 264.7 macrophage-like cells and a B-cell precursor acute lymphoblastic leukemia cell line (EU1), showing no toxicity on the former and a significant reduction of NF-κβ activation upon IL-1β stimulation [113]. Another approach consisted of the development of polymeric NPs to target the A2A adenosine receptor, as it has been demonstrated that mice with impaired adenosine signaling within macrophages of the joints develop OA spontaneously [114]. Six types of adenosine functionalized poly(lactic acid)-poly(ethylene glycol) (PLA-PEG) NPs were tested in vitro on IL-1β-stimulated macrophage-like RAW264.7 cells and in vivo in a rat post-traumatic knee OA model, which was induced through anterior cruciate ligament (ACL) rupture with a single load of tibial compression. In vitro, the adenosine PLA-PEG NPs led to a significant reduction of IL-6, MMP-3, and collagen 10 mRNA expression, whilst rats injected with the NPs into the joint showed a significant reduction in knee swelling, cartilage protection, and decreased OARSI score when compared with control rats [115].

Polymeric NPs with the ability to release the drug in response to physicochemical stimuli have been also investigated. Considering the drop of the joint’s pH in inflammatory conditions, acid-activable curcumin polymer (ACP) NPs were designed to maximize the release of curcumin in diseased joints. Curcumin is a flavonoid from natural origin with anti-inflammatory and antioxidant properties in vitro and in vivo [116]. Consequently, the ACP–curcumin NPs were tested in mice models of knee OA induced by mono iodoacetic acid, which is a corrosive molecule that destroys joint’s components and stimulates an inflammatory response [117]. The injection of ACP–curcumin NPs into the inflamed acid joints led to their fusion to form micelles and curcumin release, resulting in the suppression of TNF-α and IL-1β [118]. Hemoglobin-based PLGA-PEG-conjugated nanogenerators with the ability to produce nitric oxide (NO) upon a photothermal trigger (650 nm light irradiation) were loaded with short interfering RNA (siRNA) to inhibit Notch1. Upon the intra-articular injection of these NPs in the hindlimbs of papain OA murine models and local irradiation, the mice showed a significant reduction in the arthritis score, reduced cartilage erosion, and a lower synovial inflammation, characterized by the immunohistochemical evaluation of TNF-α, IL-1β, IL-6, and Notch-1 [119].

### 4.4. Hyaluronic Acid-Based Polymeric Nanoparticles

The local injection of HA is commonly used in the clinic, as described in Section 3. Interestingly, the use of hyaluronic acid (HA) as a free polymer, not loaded with pharmacological agents, has presented beneficial effects as a treatment for OA, due to its intrinsic immunomodulatory properties [65]. Thus, in this separate section, we provide a summary of some investigations using HA polymers to prepare NPs and their results.

Mota et al. have prepared PLGA-NPs with oleic acid and HA on their surface, showing a similar anti-inflammatory activity to diclofenac and higher than HA free solution, in rats implanted with cotton pellets subcutaneously to promote granuloma and inflammation [120]. HA and chitosan (HA/CS) NPs have also been developed to deliver into the joints the gene CrmA, cytokine response modifier A, a protease inhibitor that binds and blocks caspase-1 enzyme, which is responsible for the generation of IL-1β in OA. These HA/CS-CrmA-NPs showed a constant release of CrmA up to 70% by day 22. The intra-articular injection of these NPs in rats with knee OA showed attenuation of cartilage lesion, increase in glycosaminoglycans, intact surface, conservation of type II collagen in the joint, reduced inflammatory infiltrate, drop of IL-1β, MMP-3, and MMP-13, and better OARSI score [121]. These investigations provide satisfactory results related to the application of hyaluronic acid-based nano-approaches in OA.

### 4.5. Liposome-Based Anti-Inflammatory Approaches

Liposomes are nanospherical vesicles made up of a phospholipidic bilayer that resembles the normal cellular membrane. Due to their amphiphilic nature, liposomes can be used to deliver both hydrophilic drugs (localized in the inner core) and lipophilic drugs (trapped in the lipidic bilayer) [97]. Several liposome-based approaches have been explored to deplete synovial macrophages to prevent their detrimental effect on joint’s components. As an example, clodronate liposomes were able to ameliorate inflammation in STR/ort mice developing OA in a progressive way comparable to humans [122]. Specifically, STR/ort mice were injected intraperitoneally with clodronate liposomes to induce macrophage apoptosis, and this treatment resulted in a drop of IL-1β and TNF-α RNA expression in the synovial tissue. Interestingly, an increase in the nerve growth factor, related with persistent pain in OA patients, was also observed [78]. Similar findings were also reported in rat models of monoiodoacetate-induced OA [123]. In contrary to these results, a very recent study by Bailey et al. has demonstrated that intra-articular injection of clodronate liposomes in post-traumatic arthritis murine models did not reduce the synovitis, and the M1/M2 ratio was increased in some treated mice, advocating that clodronate liposomes could lead to the depletion not only of the M1 pro-inflammatory macrophages, but also of the M2-like immunomodulator counterparts [124]. Indeed, clodronate liposomes may present high systemic toxicity and poor targeting specificity toward the joint, thus killing macrophages in other organs, which limit the confidence on the experimental observations and also their possibilities for safe translation to the clinic. These evidences encourage higher efforts on strategies to reprogram macrophages versus their depletion.

Liposomes loaded with indomethacin were formulated using different concentrations of phosphatidyl choline, cholesterol, and stearylamine or phosphatidyl glycerol, which are common lipids that are present in the cellular plasmatic membrane. This composition resulted in prolonged and sustained release of the drug in vitro and higher anti-inflammatory activity in both acute and chronic arthritis rat models compared to the free indomethacin, as well as a reduction of gastrointestinal ulcers [125]. Similarly, liposomal formulations containing celecoxib and hyaluronate were intra-articularly injected into the knee joints of rabbits with surgically induced OA. Two weeks after the surgery, animals were sacrificed, and histological specimens from their joints demonstrated a significant alleviation of cartilage degeneration compared to celecoxib or hyaluronate alone [126]. Liposomal dexamethasone reduced the inflammatory response of human macrophages in vitro [127] and reduced local and systemic inflammation in adjuvant arthritis rat models. Liposomal dexamethasone showed prolonged effect (up to 48 h) and higher efficacy at lower doses than free drug [128]. Avnir et al. designed liposomes with a high loading and prolonged release of prednisolone hemisuccinate, an amphipathic weak acid glucocorticoid, by modulating the gradient of calcium ions concentration between the inner core of the liposomes and the medium. Consistently with their in vitro studies, the intravenous injection of the prednisolone liposomes led to a better arthritis score than the free drug in Lewis rats with adjuvant-induced arthritis [129]. Finally, hybrid nanosystems based on the conjugation of calcium phosphate NPs with liposomes decorated with folate were loaded with the immunosuppressive drug methotrexate and siRNA to inhibit NF-kβ activation in joint macrophages. Upon intravenous administration, these folate-targeted liposomes reduced significantly the paw thickness and arthritis score versus similar non-targeted liposomes in murine models of intradermal collagen-induced arthritis. Indeed, a specific macrophage uptake, related to the high expression of folate receptors, was demonstrated in vitro using LPS-activated RAW 264.7 cells [130].

### 4.6. Other Nanomaterial-Based Anti-Inflammatory Nanoparticles

Other nanomaterials, such as carbon-based NPs or metal-based NPs, have been also investigated to target macrophages in OA. For example, fullerenes are hollow nanospheres consisting of carbon atoms connected by single and double bonds with antioxidant properties [131]. The intra-articular injection of fullerenes alone or in combination with HA in surgically induced rabbit models of knee OA showed a dose-dependent fullerene therapeutic effect and synergistic activity with HA [132]. Similarly, fullerol NPs (a polyhydroxylated form of fullerene [133]) showed anti-inflammatory activity in vitro and the ability to prevent synovial inflammation in rat models of OA induced by the intra-articular injection of monoiodoacetate [134]. Carbon nanotubes or graphene-based nanomaterials have also been investigated for OA treatment [135,136,137,138,139]; however, specific studies to target macrophages are lacking. For example, Liu et al. observed a reduction of MMP-3 in the knee joints of rats following the intra-articular injection of hyaluronic acid reinforced with graphene oxide [139], and it is reasonable to speculate that this effect is secondary to macrophages’ modulation. In another report, PEGylated single-walled carbon nanotubes (SWCNTs) loaded with antisense oligomers injected in the knee joint of OA mice showed a prolonged persistence in joint cavity (more than 14 days), without induction of TNF-α or IL-1β. The loading of SWCNTs with oligomers showed the ability to inhibit gene expression in homeostatic and hypertrophic chondrocytes in vivo [137]. To our knowledge, the direct effect of this approach on macrophages was not evaluated; however, we envision the functionalization and optimization of these types of SWCNTs as an interesting strategy to target and reprogram macrophages in the joint.

Silica NPs, characterized by a good biocompatibility and high capacity to load proteins, were used to deliver hyaluronan synthase 2 toward cnidocytes in the temporomandibular joint of OA rat models with the aim to increase HA production and achieve anti-inflammatory effects. The single intra-articular injection of these silica NPs showed histological improvements lasting for more than 3 weeks, thus improving the direct HA supply, which requires multiple painful injections [140]. The encapsulation of gold NPs conjugated with fish oil proteins was also explored for OA. The injection of these hybrid nanosystems in the joint of rat models in which OA was induced using bacterial collagenase showed the ability to reduce inflammation [141]. Selenium NPs dispersed in P-coumaric acid (an anti-inflammatory compound) were applied to reduce inflammation in rats with rheumatoid arthritis, reducing the expression of pro-inflammatory activators, such as COX-2 [142]. Finally, within the group of metal organic frameworks (MOFs), zeolitic imidazolate framework-8 (ZIF-8) NPs were designed to deliver s-methylisothiourea hemisulfate salt and catalase to synovial macrophages, to inhibit iNOS and H_2_O_2_ activity. These NPs were decorated with adsorbed CD16/32 antibodies to improve their recognition by M1-like macrophages and prolong their retention into the joint [143]. Their intra-articular administration in ACLT-induced OA murine models led to significant histological improvements characterized by the narrowing reversion of joint spaces, decrease of CD16/CD32-positive M1-like macrophages, and increase of CD206-positive M2-like counterparts [143].

## 5. Conclusions

Here, we have described the pathophysiology of OA, with a particular focus on the inflammation supported by macrophages, a brief summary of the current pharmacological treatment, and a wide range of nanoparticle-based approaches to mitigate the local inflammation and improve the recovery and cure of OA disease (Figure 3). Taking this information into account, we foresee a bright future in the use of nanomedicines for the therapeutic manipulation of macrophages in the context of OA. In the last decade, we have perceived a progressive increase of experimental efforts to implement new nanotechnological strategies to improve the delivery of traditional anti-inflammatory drugs, such as NSAIDs and corticoids, toward diseased joints. Novel molecules not yet approved in the clinical practice are also being intensively investigated, including anti-inflammatory peptides or gene therapies, which will likely need the implementation of nanodrug delivery approaches to avoid their degradation and/or improve their biodistribution profile, thus allowing their effective application in vivo [110,119]. In parallel, in the field of immunology, several investigations have revealed the important role of joint macrophages in OA [144,145,146]. Macrophage polarization has been recently identified as a possible marker of prognosis [145], influencing also neighbor cells in the joint (i.e., chondrocytes). However, an important challenge to clearly dissect the role of macrophages versus other cellular and molecular components in OA is the availability and relevance of the in vitro and in vivo models commonly used in pre-clinical research. In particular, there is a lack of appropriate in vitro models of chronic inflammation mimicking the joint OA environment. The information related to the specific therapeutic targeting of macrophages in the joints is also limited; thus, if the higher precision in local macrophage targeting could lead to better results remains also to be elucidated. We have found a significant amount of scientific reports testing new anti-inflammatory approaches that resulted in a direct or indirect effect on macrophage phenotype and functions, which was correlated with improved outcomes. Despite these promising scientific observations, the clear discernment of macrophage targeting requires costly in vivo experiments focused on NP and/or drug pharmacokinetics and biodistribution at the tissular and cellular level, which we expect to see in the coming years. Finally, a few clinical trials are investigating anti-inflammatory drugs for OA treatment [31,147,148], although focused efforts examining the role of macrophages and/or the use of nanomaterial-based approaches are lacking.

In this line of research, we are currently working in the frame of the MEFISTO consortium H2020 project granted to develop a “meniscal functionalized scaffold to prevent knee osteoarthritis onset after meniscectomy”. Therefore, we expect that the imminent combination of both fields of research, namely, the better understanding of macrophages’ role in OA together with the advances in nanotechnology to control the timely and precise delivery of drugs, will provide important benefits to improve the quality of life of patients with OA.

## Figures and Tables

**Figure 1 nanomaterials-10-01562-f001:**
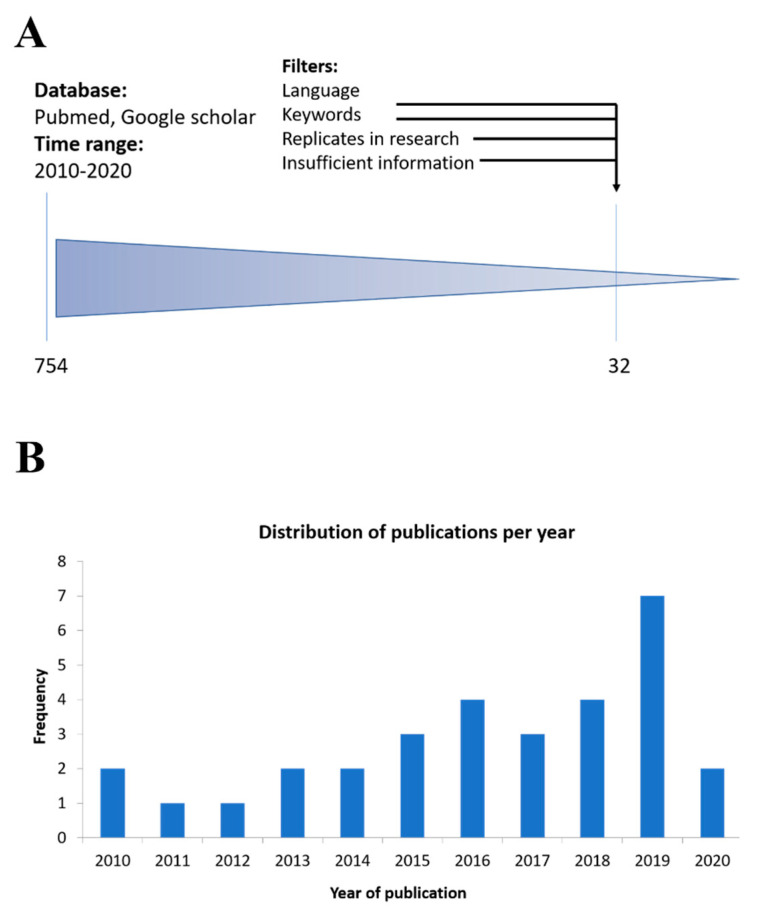
Analysis of nanomaterial-based drug delivery systems in osteoarthritis. (**A**) Procedure used for the literature survey. Studies have been retrieved using the research string “drug delivery AND osteoarthritis AND macrophages OR innate immunity OR inflammation OR immune system OR anti-inflammatory OR nanomaterials OR nanoparticles” on PubMed and Google Scholar, using as a time range the period 2010–2020. With these research criteria, we found 754 studies, and then after excluding the replicates, studies with insufficient information (unclear characteristics of the drug delivery system, missing data about drug dosage or time, uninformative anti-inflammatory readout) and studies not written in English language, we finally selected 32 studies. (**B**) Diagram showing the distribution of the included studies according to the year of publication.

**Figure 2 nanomaterials-10-01562-f002:**
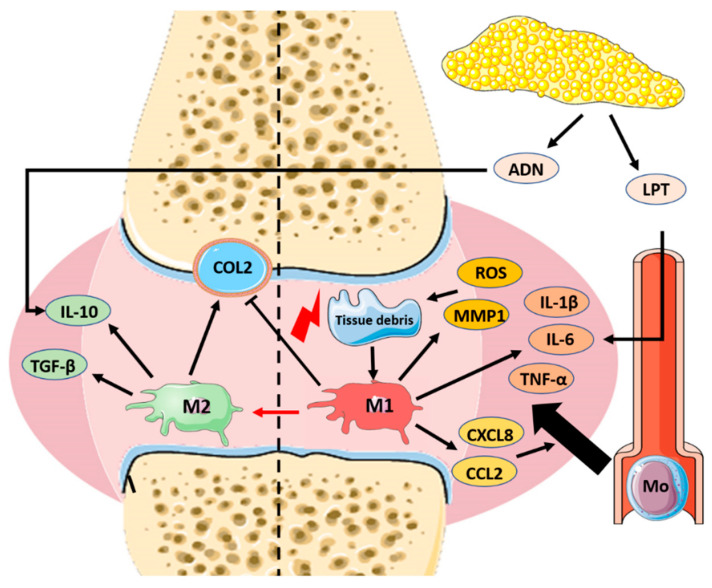
Role of macrophages in the pathophysiology of osteoarthritis. Acute trauma or chronic overuse of the joint results in the activation of resident synovial macrophages toward a harmful pro-inflammatory M1-like phenotype. Once activated, synovial M1-macrophages secrete signaling molecules, such as pro-inflammatory cytokines (interleukin (IL)-1β, tumor necrosis factor alpha (TNF-α), IL-6 and chemokines (CCL-2, CXCL-8) that lead to further leukocyte recruitment, including monocytes (Mo) that infiltrate the joint, become activated toward an M1-like phenotype, and contribute to the thickening of the synovia (a key morphological feature of synovitis). These M1 macrophages also produce matrix metalloproteinases (MMP1) and reactive oxygen species (ROS), which damage directly the joint, worsening the tissue injury and leading to a positive feedback pro-inflammatory loop. Furthermore, M1 macrophages can directly inhibit chondrogenesis through the inhibition of collagen type 2 (COL2) synthesis by mesenchymal stem cells (MSCs) in the joint. Tissue debris and molecules released from the degradation of hyaline cartilage and subchondral bone, which are related to damage of the meniscus, are responsible for the pro-inflammatory properties of the synovial fluid from the early stages of knee OA [45]. In contrast, the polarization of macrophages in the joint toward an M2-like anti-inflammatory phenotype has beneficial effects for the repair of the injury and for the cure of OA disease. It has been demonstrated that M2 macrophages are able to promote cartilage repair through the secretion of TGF-β and IL-10. Once secreted in the synovial fluid, the latter exert their pro-chondrogenic effect by stimulating chondrocytes to secrete type II collagen and proteoglycans. This beneficial action of “alternatively activated” M2 macrophages finds a clinical counterpart in a recent in vivo experiment that correlated the imbalance of M1/M2 ratio as a feature of the severity level of knee OA [10]. In addition to the local triggers of OA, the systemic immune status of the patient is a key responsible for the evolution of the disease. As examples, increased levels of systemic leptin (LPT) increase IL-6 secretion in the joint, supporting deleterious inflammation, while systemic adiponectin (ADN) has shown a protective role by upregulating the secretion of IL-10 in the articulations. Importantly, recent clinical studies have correlated the increased M1/M2 ratio with a higher severity of knee OA disease.

**Figure 3 nanomaterials-10-01562-f003:**
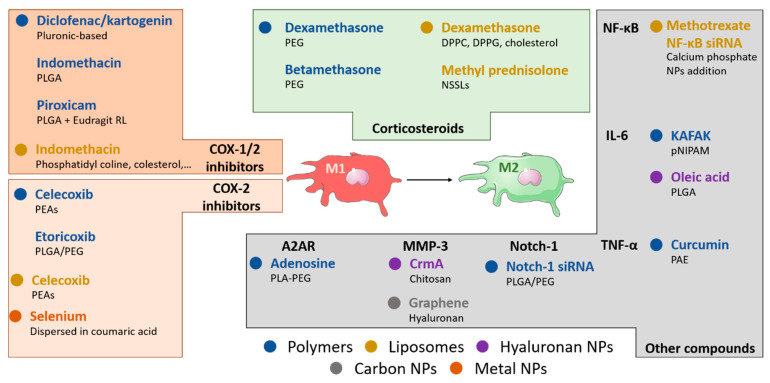
Nanotechnological approaches targeting macrophages in OA. Nanomedicine offers a wide range of drug delivery systems to inhibit the pro-inflammatory activity of M1-like macrophages in OA joints. Polymeric and hyaluronan-based nanoparticles (NPs) and liposomes have been investigated to inhibit inflammation by delivering non-steroidal anti-inflammatory drugs (NSAIDs, i.e., indomethacin, diclofenac, etc.) for COX-1/2 inhibition, COX-2 specific inhibitors (i.e., celecoxib and etoricoxib) or corticosteroids (i.e., dexamethasone, betamethasone, and prednisolone) to block glucocorticoid receptor in OA joints. NPs loaded with other pharmacological molecules have been used to selectively block the release of specific cytokines, such as IL-1β by IL-1Ra or IL-6 by KAFAK peptide. Clodronate-loaded liposomes were also used to deplete macrophages in experimental settings, showing mixed results. Metabolic or genetic manipulation of macrophages was explored using NPs to deliver adenosine and NF-κB or Notch homolog 1, translocation-associated (Notch-1) siRNAs. These approaches have showed promising experimental results related to the reprogramming of macrophages toward the M2 anti-inflammatory phenotype in the joint microenvironment, thus becoming potential candidates to improve the current treatment of OA disease.

**Table 1 nanomaterials-10-01562-t001:** Summary of nanoparticle-based approaches to target macrophages using in vivo pre-clinical models of osteoarthritis (OA).

Type of Nanoparticle	Nanocarrier Composition	Therapeutic Load	Size and Surface Charge (nm/mV)	Route of Administration	Animal Model	Therapeutic Effect	Ref
**Polymeric NPs**	Pluronic-based Thermoresponsive	*Diclofenac/Kartogenin*	305–650 nm/n.r.	i.a. (knee)	Rats	↓ of OARSI score	[98]
	Self-assembling PLGA-coated	*Indomethacin*	37–255 nm/ (−5.81)–(−9.36) mV #	i.a. (knee)	Rats	↓ of diameter; favorable hystology; ↓ TNF-α in serum	[84]
	PLGA + Eudragit RL	*Piroxicam*	221–243 nm/ (+2.4)–(+11.5) mV #	i.a. (knee)	Rats	Prolonged retention into joint compared to NPs without Eudragit RL	[99]
	PEAs	*Celecoxib*	398–836 nm/n.r. #	i.a. (knee)	Sheeps	↓ joint effusion; ↓ WBC	[101]
	PLGA/PEG	*Etoricoxib*	339 nm (mean value)/ (+1.68 ± 0.85) mV	i.a. (knee)	Rats	Favorable μCT; ↓ MMP-13 and ADAMTS-5; ↑ collagen and aggrecan	[96]
	PEG	*Dexamethasone*	130 ± 3 nm/ (−55 ± 2) mV	i.v.	Mice^†^	Accumulation in inflamed joints upon administration	[105]
	PLGA	*Betamethasone*	300–490 nm/n.r. #	i.a. (knee)	Rabbits	↓ joint swelling and temperature	[106]
	pNIPAM	*KAFAK*	238–469 nm/ (−5.38)–(−8.48) mV #	ex vivo (knee)	Bovine*	↓ IL-6	[109]
	pNIPAM/AMPS	*KAFAK*	232–358 nm/ (−6.1)–(−22.9) mV #	ex vivo (knee)	Bovine*	↓ IL-6	[110]
	PLA-PEG	*Adenosine*	129–144 nm/n.r. #	i.a. (knee)	Rats	↓ OARSI score	[115]
	Acid-activable PAE	*Curcumin*	170 nm/n.r.	i.a. (knee)	Mice	↓ TNF-α and IL-1β production	[118]
	PLGA-PEG	*NO-Hemoglobin Notch-1 siRNA*	200 nm/0 mV	i.a. (limb)	Mice	Favorable histology ↓ TNF-α, IL-6, IL-1β, Notch-1 in immunohistochemistry	[119]
**Hyaluronic acid-based NPs**	PLGA	*Oleic acid and HA*	4561 ± 3466 nm/ (−0.59)–(−16.65) mV	s.c.	Rats	↓ of inflammation in cotton pellets	[120]
	HA and Chitosan	*CrmA*	100–300 nm/n.r.	i.a. (knee)	Rats	↓ OARSI score; ↓ IL-1β, MMP-3, MMP-13; collagen conserved	[121]
**Liposomes**	Not specified	*Clodronate*	n.r.	i.p.	Mice^‡^	↓ IL-1β and TNF-α expression in synovium; ↓ NGF in the joint	[78]
	Clophosome^®^	*Clodronate*	100–500 nm/ 0 mV	i.v.	Rats	↓ IL-1β and NGF in the joint	[123]
	Phosphatidyl choline; cholesterol; stearylamine; phosphatydil glycerol	*Indomethacin*	50–100 nm/n.r. #	i.p. (knee)	Rats	↓ joint volume	[125]
	SPC and cholesterol + hyaluronan addition	*Celecoxib*	4980 nm/n.r.	i.a. (knee)	Rabbits	Favorable hystology	[126]
	DPPC + DPPG + cholesterol	*Dexamethasone*	283–310 nm/n.r.	i.v.	Rats^†^	Favorable histology and WBC count	[128]
	NSSLs	*Methyl prednisolone*	80 nm/n.r.	i.v.	Rats^†^	↓ of the arhtritis score	[129]
	Calcium phosphate NPs in liposomes	*Methotrexate* *NF-κB siRNA*	170 nm/ (−23.6) mV	i.v.	Mice	↓ limb arhtritis score ↓ paw thickness	[130]
**Carbon-based NPs**	Fullerene	*-*	1.1 nm/n.r.	i.a. (knee)	Rabbits	Favorable hystology	[132]
	Fullerol	*-*	n.r.	i.v. (knee)	Mice	Favorable hystology	[134]
	Graphene oxide	*Hyaluronan conjugation*	n.r.	i.a. (knee)	Rats	↓ MMP-3 concentration in the joint	[139]
	Carbon nanotubes	*Antisense oligomers*	109 ± 49 nm/ (−11) mV	i.a. (knee)	Mice	Inhibition of protein synthesis in chondrocytes and reduction of inflammation	[137]
**Metal-based NPs**	Silica	*Hyaluronan synthase 2*	175 nm/ (+12) mV	i.a. (TMJ)	Rats	Favorable hystology	[140]
	Gold	*Fish oil protein, both in DPPC liposomes*	15.3 ± 1.9 nm/(+4.15 ± 3.9) mV	i.a.	Rats	Reduction of inflammation	[141]
	Selenium	*NPs dispersed in coumaric acid*	68,000 ± 10,000 nm/n.r.	i.p.	Rats	Reduction of catalase, COX-2, GPx1	[142]
**Other NPs**	ZIF-8 (MOF)	*S-methylisothiourea Catalase Anti-CD16/32*	160 nm/ (−13)–(+20) mV #	i.a. (knee)	Mice	Favorable histology and X-ray	[143]

**Legend: ADAMTS-5**: A disintegrin and metalloproteinase with thrombospondin motifs 5; **AMPS:** 2-acrylamido-2-methyl-1-propanesulfonic acid; **COX-2:** cyclooxygenase 2; **CrmA:** cytokine response modifier A; **DPPC:** 1,2-dipalmitoyl-sn-glycero-3-phosphocholine; **DPPG:** 1,2-dipalmitoyl-sn-glycero-3-(phosphor-rac-(1-glycerol))(sodium salt); **GPx1:** Glutathione peroxidase 1; **HA:** hyaluronic acid; **i.a.:** intra-articular; **IL-1β:** interleukin-1β; **IL-6:** interleukin-6; **i.p.:** intraperitoneal; **i.v.:** intravenous; **μCT**: microtomography; **MMP-3:** matrix metalloproteinase 3; **MMP-13:** matrix metalloproteinase 13; **NGF:** nerve growth factor; **n.r.:** not reported; **NSSLs**: sterically stabilized nanoliposomes; **OARSI**: Osteoarthritis Research Society International; **PAE:** poly(β-amino ester); **PEAs:** poly(ester amide)s; **PEG:** poly(ethylene glycol); **PLA:** poly(lactic acid): **PLGA**: poly(lactic-*co*-glycolic acid); **pNIPAM:** poly(N-isopropylacrylamide); **s.c.:** subcutaneous; **SPC:** soybean phosphatidylcholine; **TMJ:**
*temporomandibular joint**;***
**TNF-α:** tumor necrosis factor alpha; **WBC:** white blood cells; **ZIF-8**: zeolitic imidazolate framework-8; ***** Ex vivo; † Rheumatoid arthritis**; ‡** STR/Ort mice that develops spontaneously OA. **#** NPs with different but defined size and charge within the indicated range were tested.

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
