# Peer review of "Therapeutic Manipulation of Macrophages Using Nanotechnological Approaches for the Treatment of Osteoarthritis"

_nanomaterials, 2020, doi:10.3390/nano10081562_

Round 1
Reviewer 1 Report
This review article deal with a nanotechnologies for treament of osteoarthritis by manipulation of macrophages. The manuscrip was organized depending on the used materials to prepare nano-sized carrier. Before the futhure consideration, few questions are must be addressed.
- The subtitles are arranged by the materials. However, hyaluronic acid is also kind of polymer. Is there a reason for separate the HA from polymer? And author have to mention the importance of HA for the treatment of OA in the appropriate paragraph.
- This review deals with a nanotechnology based materials for OA treatment. Therefore, author have to summarize the feature of developed nanomaterials such as size and surface charge, etc. in Table 1.
- The all surmmaized technologies in this review are just targeted to macrophage or the other component such as collagen? There are many approaches to treat OA in these days. Therefore, author have to mention the purpose of the summarized nanomaterials and their target.
Reviewer 2 Report
This is a timely review of therapeutic manipulations of macrophages for treatment of osteoarthritis. It is generally well written. However, a major concern is that, although the topic is manipulations of macrophages, the cited studies are manipulations of the pro-inflammatory pathways in the immune system involving multiple immune cells. To overcome this deficiency, the authors need to address the following concerns.
1) The role of inflammation in OA pathogenesis has been well documented in the literature. This review article include all recent studies of inhibition of inflammation in OA models. This needs to be more focused on the studies that target macrophages rather than the general inhibition of inflammation. Some of these study descriptions can be shortened.
2) The review article should include the studies (clinical, animal models, and basic research) that address the role of macrophages in OA pathogenesis. This is the basis of macrophage manipulation for OA treatment.
3) The review article should include strategies for macrophage-targeted delivery. Although such studies may be few for the treatment of OA so far, the authors may want to cite macrophage targeted delivery in other fields. These may be useful to designing macrophage manipulation experiments in the OA field.
